

# TIN-X version 3: update with expanded dataset and modernized architecture for enhanced illumination of understudied targets

Vincent T. Metzger[1], Daniel C. Cannon[2], Jeremy J. Yang[1],
Stephen L. Mathias[1], Cristian G. Bologa[1], Anna Waller[3],
Stephan C. Schürer[4], Dušica Vidović[4], Keith J. Kelleher[5],
Timothy K. Sheils[5], Lars Juhl Jensen[6], Christophe G. Lambert[1],
Tudor I. Oprea[1] and Jeremy S. Edwards[1,7]

[1] Translational Informatics Division, Department of Internal Medicine, University of New Mexico Health Sciences Center, Albuquerque, New Mexico, United States
[2] Elevato Digital, Columbia, Missouri, United States
[3] Center for Molecular Discovery, University of New Mexico Comprehensive Cancer Center, Albuquerque, New Mexico, United States
[4] Department of Molecular and Cellular Pharmacology, Miller School of Medicine, University of Miami, Miami, Florida, United States
[5] National Center for Advancing Translational Science, Rockville, Maryland, United States
[6] Novo Nordisk Foundation Center for Protein Research, Faculty of Health and Medical Sciences, University of Copenhagen, Copenhagen, Denmark
[7] Department of Chemistry and Chemical Biology, University of New Mexico, Albuquerque, New Mexico, United States

Corresponding author
Jeremy S. Edwards,
jsedward@unm.edu

## ABSTRACT

TIN-X (Target Importance and Novelty eXplorer) is an interactive visualization tool for illuminating associations between diseases and potential drug targets and is publicly available at newdrugtargets.org. TIN-X uses natural language processing to identify disease and protein mentions within PubMed content using previously published tools for named entity recognition (NER) of gene/protein and disease names. Target data is obtained from the Target Central Resource Database (TCRD). Two important metrics, novelty and importance, are computed from this data and when plotted as log(importance) *vs.* log(novelty), aid the user in visually exploring the novelty of drug targets and their associated importance to diseases. TIN-X Version 3.0 has been significantly improved with an expanded dataset, modernized architecture including a REST API, and an improved user interface (UI). The dataset has been expanded to include not only PubMed publication titles and abstracts, but also full-text articles when available. This results in approximately 9-fold more target/disease associations compared to previous versions of TIN-X. Additionally, the TIN-X database containing this expanded dataset is now hosted in the cloud *via* Amazon RDS. Recent enhancements to the UI focuses on making it more intuitive for users to find diseases or drug targets of interest while providing a new, sortable table-view mode to accompany the existing plot-view mode. UI improvements also help the user browse the associated PubMed publications to explore and understand the basis of TIN-X's predicted association between a specific disease and a target of interest. While implementing these upgrades, computational resources are balanced

between the webserver and the user's web browser to achieve adequate performance while accommodating the expanded dataset. Together, these advances aim to extend the duration that users can benefit from TIN-X while providing both an expanded dataset and new features that researchers can use to better illuminate understudied proteins.

## INTRODUCTION

The rapid growth of research publications has exceeded the limits of human processing (*Hunter & Cohen, 2006*) and this is especially the case for PubMed. PubMed is a widely-used database of biomedical literature, containing millions of articles on various topics related to medicine and life sciences. One of the challenges in effectively using PubMed is the vast amount of information available which can make it difficult to extract relevant information efficiently, especially when exploring understudied drug targets. named entity recognition (NER), is a type of natural language processing that can help to address this challenge (*Grishman & Sundheim, 1996*; *Liu, Chen & Xia, 2022*). NER is a technique that involves identifying and extracting named entities, such as genes, proteins, diseases, drugs, and other biomedical concepts, from publication text data. By applying NER to PubMed articles, the Jensen Lab leads an important effort to regularly extract valuable information about biomedical concepts mentioned in the literature (*Pletscher-Frankild et al., 2015*; *Grissa et al., 2022*). These data can help to increase our knowledge of various biomedical topics, including disease mechanisms, drug targets, and therapeutic interventions.

In response to the growing body of peer-reviewed biomedical literature, a new tool called Target Importance and Novelty eXplorer (TIN-X) was introduced in 2017 (*Cannon et al., 2017*). TIN-X is a featured resource of the Illuminating the Druggable Genome (IDG) consortium. TIN-X is a public web application used to identify, visualize, and explore target/disease associations (*Cannon et al., 2017*). TIN-X seeks to leverage the extensive PubMed NER data made available by the Center for Protein Research (CPR), along with the Target Central Resource Database (*Sheils et al., 2021*; *Kelleher et al., 2023*) (TCRD) to facilitate the illumination of understudied proteins while also serving as a powerful tool for advancing biomedical research and improving human health. TIN-X has been continuously available at https://newdrugtargets.org since originally released in 2017 (*Cannon et al., 2017*). Source code for the API and UI are available at https://github.com/unmtransinfo/tinx-api and https://github.com/unmtransinfo/tinx-ui, respectively.

Since its launch, TIN-X has played an important role inspiring and aiding research into understudied targets. TIN-X has been recently cited in studies seeking to illuminate specific targets, such as the "orphan" (no known biological ligand or physiological
function) G Protein-Coupled Receptor (GPCR) named *GPRC5B*, which was reported to be critical for lymphatic development (*Xu et al., 2022*). Specifically, the authors used TIN-X to explore the association between various GPCRs and lymphedema. While this study focused on *GPRC5B*, because it has the highest expression levels in zebrafish and mouse models, the investigators also illuminate several other orphan receptors: *AGDRF5/ GPR116*, *FZD8*, and *GPR61* (*Xu et al., 2022*). A different study (*Pirola & Sookoian, 2021*), used TIN-X to explore the lipidome in metabolic dysfunction-associated steatotic liver disease (MASLD). The authors visualized targets associated with fatty liver disease on an Importance *vs.* Novelty plot, combined with implementation of an integrated pathway-level analysis of genes and lipid-related metabolites associated with MASLD (*Pirola & Sookoian, 2021*).

TIN-X is one of several online tools which rely upon text-mining and NER of the PubMed *corpus*, thereby adding value to this indispensable resource. The NIH NLM PubTator (*Wei, Kao & Lu, 2013*) system provides NER *via* public web app and API, for a variety of biomedical entity types including diseases and genes. The DISEASES database (*Grissa et al., 2022*), likewise infers disease-gene associations, from the same NER pipelines also used for TIN-X. These are two examples, but there are many others, as text-mining and domain-and vocabulary-aware NER have become standard tools of data science. However, TIN-X is unique by virtue of its two bibliometric measures, Novelty and Importance, and their combined use and interactive visualization, for rapid exploration and prioritization in the service of IDG use cases.

While the TIN-X web application at https://newdrugtargets.org has been used for years as a powerful tool for exploring target/disease relationships, it became evident that further innovation is required to satisfy the needs of existing TIN-X users while simultaneously attracting new users to the software. For example, users of the previous version of TIN-X described the relative difficulty of searching through scatterplot points for specific targets or diseases of interest; this shortcoming is now addressed by the creation of the new table-view mode. The effort to revitalize and improve TIN-X required considerable troubleshooting, bug fixes, software dependency upgrades, and the implementation of user-recommended improvements. This work has culminated in the creation and public release of the new TIN-X version 3.0. Here we introduce the new version of TIN-X with a significantly expanded dataset, modernized architecture, a REST API, an open-source repository, a database in the cloud, and new UI features.

## MATERIALS AND METHODS

Two important metrics, novelty and importance, are computed from this data, and when plotted as log(importance) *vs.* log(novelty), aid the user in visually exploring the novelty of drug targets and their associated importance to diseases. Bibliometric statistics including the derived Novelty ($N_i$) and Importance ($I_{ij}$) scores are computed using the equations below, for target ($i$) and disease ($j$), where $T_k$ and $D_k$ are the numbers of targets ($T$) and diseases ($D$) in abstract ($k$). A greater importance score implies that more has been published about the association between the given target and the selected disease. A greater novelty score implies that less has been published about the target.

$$N_i = 1/\sum_k \frac{1}{T_k} \tag{1}$$

$$I_{ij} = \sum_k \frac{1}{T_k \cdot D_k} \tag{2}$$

In addition to being a public web application, TIN-X is now an informatics workflow, and a REST API. As depicted in Fig. 1, users can access the TIN-X user interface (UI) at https://newdrugtargets.org. Unlike the original version of TIN-X, the UI and the REST API are now separate components, as depicted in Fig. 1. User activity on the TIN-X public web application results in API requests which in turn access the TIN-X database. Alternatively, users can query the data directly *via* the REST API (https://api.newdrugtargets.org/), which is supported by Swagger documentation. The TIN-X database has been upgraded from an instance of mySQL v.8 to Amazon Relational Database Service (RDS). Together, the TIN-X UI, API, and database are all hosted in the cloud using Amazon Web Services (AWS). The TIN-X database relies on TCRD for target data and the DISEASES 2.0 resource (*Pletscher-Frankild et al., 2015*) (*Grissa et al., 2022*) for text-mined PubMed content. In Fig. 1, Blue arrows depict the flow of data from these two important sources.

TIN-X relies on the DISEASES (*Grissa et al., 2022*) resource for text-mined PubMed associations which are updated weekly by the Jensen Lab. For text-mined associations, the number of disease-gene associations increased by at least nine-fold at all confidence cutoffs (*Grissa et al., 2022*) compared to previous versions of the DISEASES 2.0 resource. This was achieved primarily by adding full-text articles to the collection in addition to the titles and abstracts previously included. To a lesser extent, improvements to the disease and the gene dictionaries that are used for Named Entity Recognition helped increase the number of target/disease associations (*Grissa et al., 2022*).

## New in version 3.0

The most useful new feature in the improved UI is the creation of a switch allowing users to toggle between a plot (see Fig. 1) and a newly created table viewing mode (Figs. 2A, 2B). We added the new table view mode for both the *Browse Diseases* (Fig. 3A) and *Browse Targets* (Fig. 3B) TIN-X operational modes. This new feature allows users to view, sort, filter-search, and otherwise interact with all the information present in the log(Novelty) *vs.* log (Importance) scatterplots in a tabular format. In previous versions of TIN-X, this important target or disease data was not simultaneously accessible to the user in the scatterplot view as it now exists in the new table view mode. TIN-X Plot-view users can sort the table results ascending or descending by any of the fields and they can also perform a filter-search for results that immediately eliminates non-matching records while the user types the desired filter string. Users can click on any row in the new table view and bring up the same detailed popup that appears when a user clicks on a datapoint in the established plot view. Additionally, we have reformatted the multi-publication view to have consistent styling and to display the date of publication for each article. A browser compatibility issue was addressed regarding the slider located in the upper-right of the plot view (Fig. 2A)

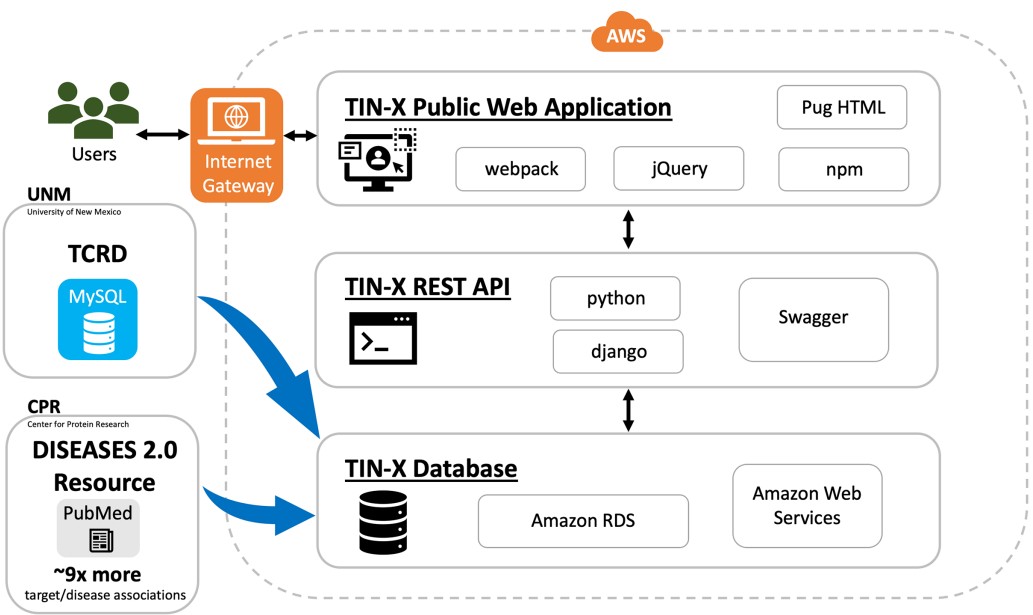

**Figure 1** **TIN-X informatics workflow.** Schematic showing data sources and the flow of information within the TIN-X informatics workflow. Users access the TIN-X user interface (UI) *via* a web browser. Unlike earlier versions of TIN-X, the UI and the REST API are separate components. User activity on the TIN-X public web application results in API requests which in turn access the TIN-X database. Users can query the data directly *via* the REST API, which is supported by Swagger documentation. The TIN-X Database has been upgraded from an instance of mySQL to Amazon RDS. Together, the TIN-X UI, API, and database are all hosted in the cloud using Amazon Web Services. Importantly, the TIN-X Database relies on TCRD for target data and the JensenLab DISEASES resource for text-mined PubMed content. Blue arrows depict the flow of data from these two major sources to the TIN-X Database. Several specific web technologies are highlighted adjacent to each major component of the TIN-X application.

which is used to adjust the number of results to display. Also, when a user selects and deselects filter values while browsing results, the corresponding points in the scatterplot or table immediately disappear or appear based on the filter values being applied (Fig. 2A). Several other web browser compatibility issues and formatting inconsistencies were identified in the previous version of TIN-X and were subsequently corrected to improve the user experience. The UI has been tested with modern desktop versions of Google Chrome, Mozilla Firefox, Safari, IE/Edge, and Opera. Finally, the "About" section of TIN-X has been updated to describe version 3.0 of TIN-X with details such as when data sources were last updated and a list of contributors.

This new version of TIN-X features improvements to the TIN-X database. The TIN-X database now runs on Amazon RDS instead of mySQL v8 (*Cannon et al., 2017*). Additionally, changes were made to the TIN-X metadata table to improve performance. We also have a significantly expanded database due to Jensen Lab DISEASES resource version 2.0 indexing full-text articles from PubMed when available (*Grissa et al., 2022*) instead of only titles and abstracts. There are approximately nine-fold more target/disease associations in the new DISEASES 2.0 resource compared to previous versions
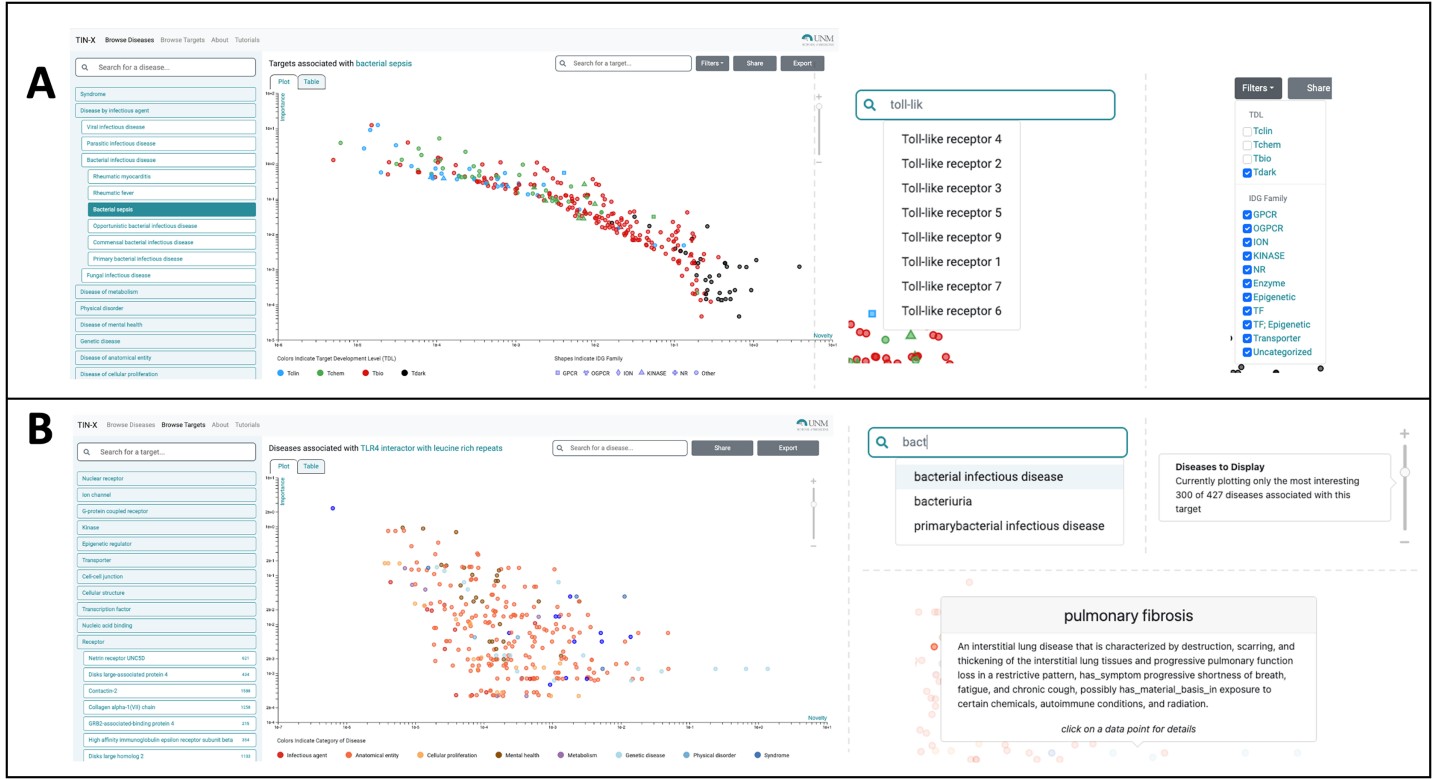

**Figure 2 Using TIN-X 3.0: browsing by disease or by target.** (A) Users can Browse Diseases by exploring the nested drop-down menus on the left or by typing the name of the disease of interest into the search/filter field in the top left. In the main plot area, each point is a Target associated with the disease selected in this example (bacterial infectious disease). These points are colored based on Target Development Level (TDL), and the shape of each point indicates the IDG Family, as indicated on the legend at the bottom. The next screenshot shows a user searching for a specific Target among the many that exist. The autocomplete suggests various Toll-like receptor targets, which if selected, highlight the corresponding point on the plot. In the upper-right, Filters can be applied to add or remove points from the plot based on TDL category and/or IDG Family. (B) In Browse Targets mode, users can search for a target of interest by using the autocomplete search or by browsing through the menus. The main plot area depicts each associated disease and is colored based on the category of disease. The next screenshots on the right show the use of the autocomplete search for diseases among the results shown in the plot. Hovering over a point reveals details about the disease of interest, in this example pulmonary fibrosis. The slider on the far right of this panel appears in the upper right portion of the scatterplot view. Manipulating the slider either increases or decreases the number of associated diseases that are shown, with the default being 300. By default, only the 300 most-interesting articles (according to NDS rankings) are displayed, however the user is free to adjust the number of results plotted using this slider.

(*Grissa et al., 2022*). This represents a meaningful increase in text-mined associations that improves the chances of biomedical researchers illuminating understudied targets.

The TIN-X API is accessible at api.newdrugtargets.org and is supported by Swagger documentation. The API has been adapted to support interaction with the new TIN-X Database in the cloud (Amazon Relational Database Service). Additionally, the entire API was upgraded from Python2 to Python3 in order to improve the reliability, stability, and interoperability of this resource and to extend the useful duration of the entire TIN-X application. The TIN-X API has been optimized to accommodate the expanded dataset. Like other components of TIN-X, several bugs that had been identified in the API are now fixed. A TIN-X REST API has also been integrated with the Common Fund Data Ecosystem (CFDE) Gene Pages Partnership Project.

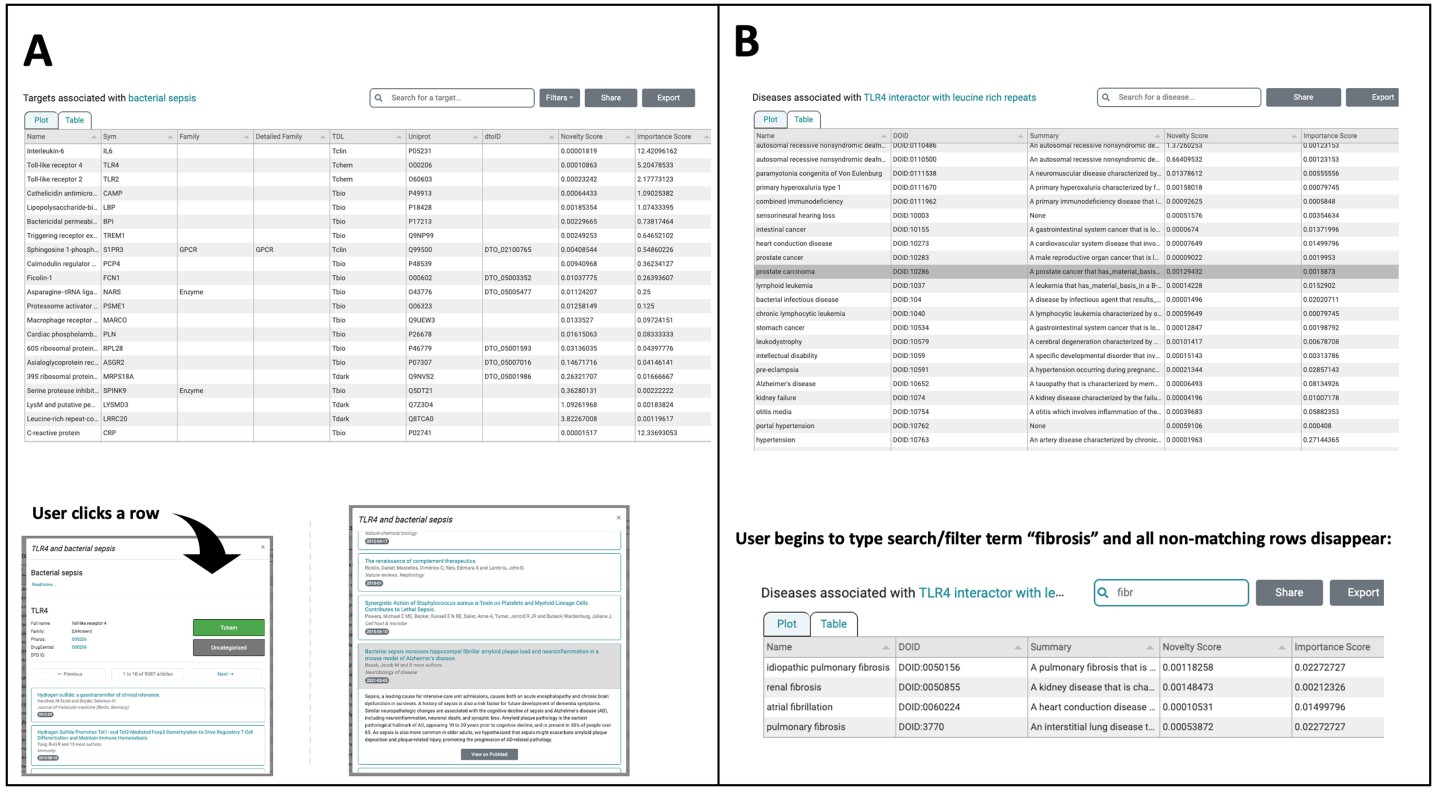

**Figure 3 The TIN-X UI now has a new plot-view to browse targets or diseases.** (A) In addition to the plot-view, the new TIN-X UI now features a table view that contains all the same information as the corresponding plot view. Users can toggle back and forth between the table view and plot view. Since this is *Browse Diseases* mode, the table shows the associated targets. All of the fields in the table are sortable and when a user clicks on a row, the detailed view appears (lower left). The screenshot in the lower right shows how a user can explore TIN-X's predicted association between the TLR-4 receptor and bacterial sepsis. Clicking on a publication row in the list shows the abstract and presents the user with an external link to access the article on PubMed. (B) This panel shows the new table-view feature within the Browse Targets mode, where associated diseases can be sorted ascending or descending by any of the fields. In both *Browse Targets* mode (shown) and *Browse Disease* mode, the User can filter the table contents by beginning to type search/filter strings into the Search field, and all non-matching rows disappear. This feature is useful for finding one desired target or disease among the many results in the table. Like the *Browse Diseases* mode in Panel A, clicking on a row in *Browse Targets* mode reveals details about the target along with associated PubMed publications.

In addition to existing as a standalone web application at https://newdrugtargets.org, TIN-X is now a featured resource within Pharos (https://pharos.nih.gov/), an NIH-hosted comprehensive drug target database and platform for exploring the druggable genome (*Kelleher et al., 2023*). As depicted in Fig. 4, text-mined target-disease associations from TIN-X are shown in an interactive scatterplot next to a circular treemap. The circular treemap groups the associations based on the hierarchy defined by the Disease Ontology (*Schriml et al., 2022*). Selecting a circle (group of diseases) or a point (individual disease) in the right panel highlights corresponding points on the scatterplot. The scatterplot is a log (importance) *vs.* log(novelty) plot, just like those produced by the standalone TIN-X web application. This successful integration of TIN-X with Pharos will help introduce Pharos users to the capabilities of TIN-X while providing an inline tool to visually explore disease associations.

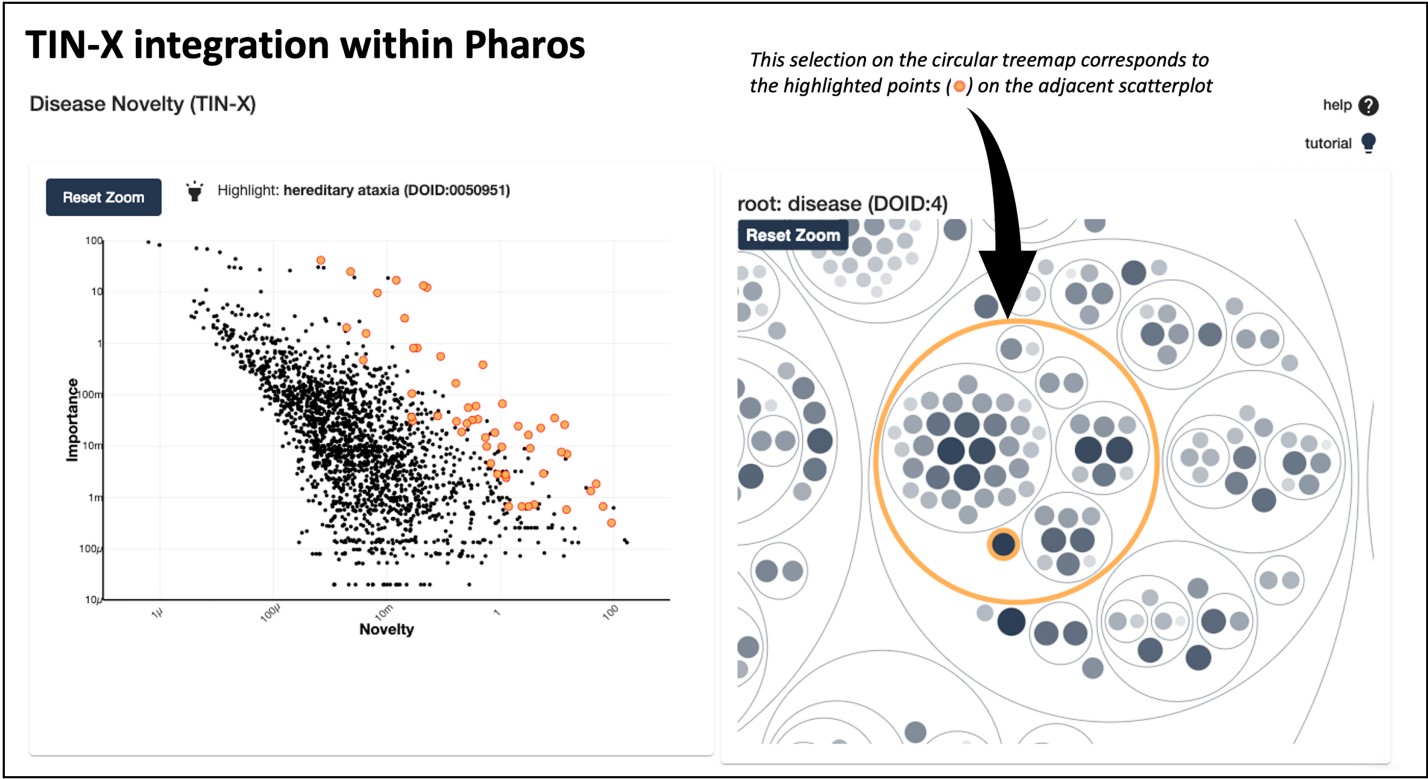

**Figure 4  TIN-X integration with pharos.** This screenshot shows TIN-X integration within Pharos. On the left, text-mined target-disease associations from TIN-X are shown in an interactive scatterplot next to the circular treemap. This example depicts the relationship between *CACNA1A* and hereditary ataxias. The circular treemap groups the associations based on the hierarchy defined by Disease Ontology. Selecting a circle (group of diseases) or a point (individual disease) in the right panel highlights corresponding points in the scatterplot on the left.

## Using TIN-X version 3.0

Users can *Browse Diseases* or *Browse Targets* by exploring the nested drop-down menus on the left or by typing the name of the target or disease of interest into the search/filter field in the top left. In the main plot area, each point represents a target associated with the disease selected in this example, bacterial infectious disease (Fig. 2A). These points are colored based on Target Development Level (TDL), and the shape of each point indicates the IDG Family, as indicated on the legend at the bottom (Fig. 2A). The next screenshot (Fig. 2A) shows a user searching for a specific Target among the many that exist. The autocomplete in this example suggests various Toll-like receptor targets, which if selected, highlight the corresponding point on the plot. In the upper-right, filters can be applied to add or remove points from the plot based on TDL category and/or IDG Family. The slider on the far right of this panel (Fig. 2A) appears in the upper right portion of the scatterplot view. Manipulating the slider either increases or decreases the number of associated diseases (or targets) that are shown. We order TIN-X target or disease results by their non-dominated solution (NDS) ranking optimizing both novelty and importance. By definition, NDS rank-one is assigned to solutions for which no solutions exist that are superior in all variables, NDS rank-two are solutions that satisfy that condition if rank-one
solutions are removed, *etc*. By default, only the 300 most "interesting" results are displayed, based on NDS ranking. However, the user is free to adjust the number of results plotted using the slider depicted in Fig. 2A. In addition to the plot view, the new TIN-X introduces a powerful table view mode. The table displays all the same data available *via* the scatterplot but has advanced filtering capabilities, the ability to sort table results ascending or descending by any of the table's fields, and a search filter that dynamically updates the displayed table results to exclude any rows that do not match the user-provided search string. Importantly, users can access the full details by clicking on any row (Fig. 3A), as was previously available by clicking on a point in the plot view. The expanded view contains details about the target and disease, an external link to Pharos, DrugCentral, a DTO ID, and the target's family classification and TDL. This view also contains a list of the PubMed publications that form the basis for the predicted association between the target and disease of interest (Fig. 3A). Clicking the title of a publication in this list reveals the publication's abstract, along with an external link to open the research article in PubMed.

Users can toggle back and forth between plot and table viewing modes as they explore the *Browse Targets* and *Browse Diseases* tools. Users can also generate a shareable link by using the Share option in the upper-right (Fig. 2A), and they can export a comma separated value (CSV) file of all the Targets or Diseases that are currently selected. In order to control the number of targets or diseases that are returned as results, the user can manipulate a slider control located in the upper right corner of the scatterplot. Clicking the "+" or "−" buttons increases or decreases the number of results. By default, all results are shown if there are less than 300 total, but if greater than 300 results exist, only the 300 most interesting results (those with the highest ranking NDS scores) are shown.

## Use case: illumination of an understudied target for Parkinson's disease

Parkinson's disease (PD) is a chronic, degenerative disorder of the central nervous system primarily affecting the motor system (*Kalia & Lang, 2015*). Both environmental and genetic factors (*Warner & Schapira, 2003*; *Kalia & Lang, 2015*; *Lill, 2016*) likely contribute to the development of PD in an individual. PD is a complex and intensely studied disease that resists more traditional, manual techniques for conducting relevant literature searches due to the vast and growing body of peer-reviewed PD publications. TIN-X is well suited to discover and explore PD-associated targets with an emphasis on illuminating understudied targets.

The *Browse Diseases* mode of TIN-X is used to explore understudied targets associated with PD. The list of diseases on the left navigation of TIN-X or the search option can be used to locate Parkinson's Disease. The "Parkinson's Disease" category contains three subcategories in the TIN-X *Browse Diseases* mode: Late-onset PD, Early-onset PD, and Juvenile-onset PD. Selecting the parent term, "Parkinson's disease" in the diseases navigation shows results for all three forms of PD. Figure 5A shows the resulting table view of targets associated with PD, by default ordered by the NDS rank of Importance and Novelty scores. Examination of this table (Fig. 5A) reveals several T_dark targets among the top targets ordered by increasing nds_rank.

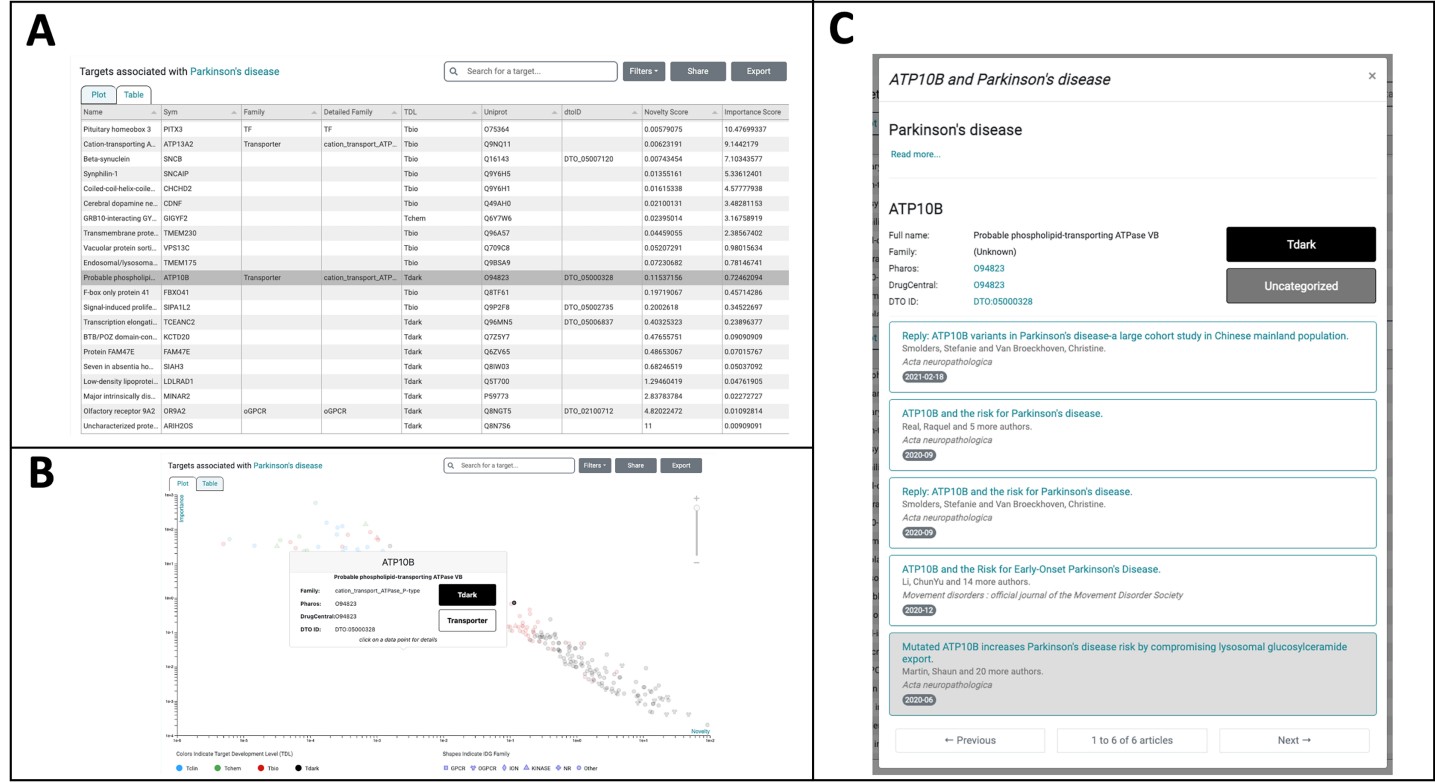

**Figure 5 TIN-X Use-case: Parkinson's disease and *ATP10B*.** (A) Targets associated with Parkinson's disease (PD) are depicted in Table-View mode and are initially ordered by the NDS ranking of importance and novelty scores. The highlighted row corresponds to the understudied T_dark target *ATP10B*, which is the subject of further exploration in TIN-X. (B) Targets associated with PD are shown in Plot-view mode on a log (Importance) *vs.* log(Novelty) scatterplot, with colors indicating Target Development Level (TDL) and shapes corresponding to IDG families. Hovering over a target of interest, *ATP10B*, reveals further details about this target. (C) Clicking on the *ATP10B* datapoint in the TIN-X scatterplot or, alternatively, clicking the *ATP10B* row in the table brings up the detailed view of *ATP10B* and PD. The articles responsible for TIN-X's predicted association between *ATP10B* and PD are displayed, with the article of interest highlighted.     

Among the top-ranked T_dark targets (19th overall in the table) is the T_dark target *ATP10B*, which TIN-X predicts to have high PD-relevance. The highlighted row in Fig. 5A is the understudied target *ATP10B*, which encodes a late endosomal lipid flippase (*Timcenko et al., 2019*). Figure 5B depicts the same targets in Plot mode on a log (Importance) *vs.* log(Novelty) scatterplot with the T_dark target *ATP10B* highlighted *via* mouseover. Clicking on this datapoint in the scatterplot (or alternatively clicking the corresponding row in the Table-view) reveals details about *ATP10B* and PD including external links to Pharos and DrugCentral as shown in Fig. 5C. The six articles responsible for TIN-X's predicted association between *ATP10B* and PD are listed here (Fig. 5C).

A noteworthy article in this list proposes that mutated forms of ATP10B increases the risk of PD by compromising lysosomal glucosylceramide export (*Martin et al., 2020*). This is the first study to demonstrate that *ATP10B* encodes a late endosomal lipid flippase which is responsible for translocating glucosylceramide and phosphatidylcholine from the luminal to the cytosolic membrane leaflet. The authors also conducted genetic screening among cohorts of patients with early-onset PD and dementia with Lewy bodies and identified seven total patients as heterozygous mutation-carriers of the *ATP10B* P4-Type

ATPase gene (*Martin et al., 2020*). They show in isolated neurons that the loss of *ATP10B* is responsible for lysosomal dysfunction and associated cell death which is important because lysosomal dysfunction is known to be involved in PD pathology. The results of this study suggest that "recessive loss-of-function mutations in ATP10B increase the risk for PD *via* disturbed lysosomal export of glucosylceramide and phosphatidylcholine" (*Martin et al., 2020*) Specifically, the authors make a case that loss-of-function mutations in *ATP10B* lead to reduced lysosomal concentrations of glucosylceramide which in turn causes lysosomal dysfunction. In this example, TIN-X leads the user to this significant recent research, which proposes specific mechanisms by which an understudied target, *ATP10B*, might be implicated in PD.

Other articles and associated correspondence are listed among the TIN-X article results for ATP10B and PD, including: "*ATP10B* variants in Parkinson's disease: a large cohort study in Chinese mainland population" (*Zhao et al., 2021*), and also the correspondence "Reply: *ATP10B* variants in Parkinson's disease: a large cohort study in Chinese mainland population" (*Smolders & Van Broeckhoven, 2021*). The authors of this article (*Zhao et al., 2021*) and associated correspondence (*Smolders & Van Broeckhoven, 2021*) were motivated by (*Martin et al., 2020*) but did not find convincing statistical evidence among the cohorts they examined linking mutations in *ATP10B* to PD. However, due to limitations in these population studies, further investigation is needed to better assess the relationship between genetic mutations in *ATP10B* and PD. As shown in Fig. 5C, the addition of article publication dates and improved formatting in Version 3.0 of TIN-X helps the user explore the collection of articles responsible for the predicted association between *ATP10B* and PD. In this case, the chronology of associated articles is important because the later studies (*Smolders & Van Broeckhoven, 2021*; *Zhao et al., 2021*) cite the earlier, groundbreaking work linking ATP10B and PD (*Martin et al., 2020*). Users exploring the list of articles can rapidly understand the basis of TIN-X's prediction by clicking on an article title, revealing the abstract for that article within the TIN-X UI and providing an external link to the full-text article.

In this use-case, TIN-X is shown to aid in illuminating the understudied target *ATP10B* by assigning high Importance and Novelty scores to this T_dark target while providing a convenient interface for exploring the research articles responsible for this predicted association.

## CONCLUSIONS

TIN-X version 3.0 is much improved compared to its predecessor because of the expanded dataset, REST API, and the introduction of cloud storage for the TIN-X database. By analyzing large volumes of scientific literature, TIN-X can be used by researchers to identify patterns and relationships between targets and diseases that might not otherwise be apparent. Its novelty *vs.* importance visualizations, accompanied by the new table view mode, can help researchers to identify potential targets for drug discovery or to develop new insights into the underlying mechanisms of a particular disease. TIN-X directly benefits from major improvements to the DISEASES 2.0 resource which has been expanded to include NER indexing of full-text open-access articles when available. This

growth in the quantity of DISEASES 2.0 content is accompanied by ongoing efforts to improve the quality of indexed articles by excluding content from paper mills (*Grissa et al., 2022*). The improvements featured in Version 3.0 of TIN-X make it a more capable tool for exploring relationships between targets and diseases, especially understudied targets which often have relatively few publications linking them to the disease of interest. The dataset expansion and the corresponding ~9-fold increase in the total number of target/disease associations enhances the illumination of understudied, T_dark targets by presenting the user with a greater number of relevant research articles to explore.

A limitation of automated bibliometry is that it relies on the availability and quality of text. TIN-X can only analyze the data that is available to it, so TIN-X may miss some important research that has not been published in scientific journals, or that has been published but not yet included in TIN-X. Importantly, TIN-X can only access the abstract (and not the full-text) for journal articles that are not open-access. Another limitation to TIN-X is the occasional false or misleading association predicted between a target and a disease, as shown in the use-case where a spurious association between the top ranked T_dark target *Glycophorin-E* and Parkinson's disease was encountered (Fig. 5A). Additionally, the accuracy of TIN-X's predicted results depends on the quality of the underlying NER data. Because TIN-X depends upon TCRD and DISEASES 2.0, TIN-X is positioned to benefit from improvements to either resource. For the same reasons, TIN-X is also vulnerable to a hypothetical disruption in the biomedical data pipeline depicted in Fig. 1. Also, because the Novelty and Importance scores are precomputed to achieve reasonable runtime performance, each data update for TIN-X must be accompanied by recalculation of these values and regeneration of the TIN-X tables, presenting a potential obstacle for achieving frequent TIN-X data updates.

Future work on TIN-X will focus on improving the process of updating data from both TCRD and the DISEASES 2.0 resource. Specifically, automation of part or all of each process is expected to improve TIN-X substantially by supporting an increased frequency of data updates. Ultimately, improvements in the ability to update TIN-X data will benefit users who will have access to more recent PubMed content *via* DISEASES 2.0, plus the latest changes and updates to target information reflected in TCRD. Version 3.0 of TIN-X, featuring an expanded dataset and modernized architecture, is beneficial for researchers who are looking to identify new areas of biomedical research to explore or who want to understand the current state of a particular field. We demonstrate that TIN-X is a relevant tool for illuminating understudied drug targets and we encourage biomedical scientists to take advantage of this upgraded and improved software tool available at https://newdrugtargets.org/.

### Funding

This work was supported by the US National Institutes of Health (grants CA224370 and U24TR002278), Illuminating the Druggable Genome Knowledge Management Center (IDG KMC), Illuminating the Druggable Genome-Common Fund Data Ecosystem (IDG-

CFDE grant number OT2OD030546) and by the Novo Nordisk Foundation (grant number NNF14CC0001). The funders had no role in study design, data collection and analysis, decision to publish, or preparation of the manuscript.

## Grant Disclosures
The following grant information was disclosed by the authors:
US National Institutes of Health: CA224370 and U24TR002278.
Druggable Genome Knowledge Management Center (IDG KMC).
Druggable Genome-Common Fund Data Ecosystem (IDG-CFDE): OT2OD030546.
Novo Nordisk Foundation: NNF14CC0001.

## Competing Interests
Daniel C. Cannon is employed by Elevato Digital.

## Author Contributions

- Vincent T. Metzger conceived and designed the experiments, performed the experiments, analyzed the data, prepared figures and/or tables, authored or reviewed drafts of the article, and approved the final draft.
- Daniel C. Cannon conceived and designed the experiments, performed the experiments, analyzed the data, authored or reviewed drafts of the article, and approved the final draft.
- Jeremy J. Yang conceived and designed the experiments, performed the experiments, analyzed the data, authored or reviewed drafts of the article, and approved the final draft.
- Stephen L. Mathias performed the experiments, authored or reviewed drafts of the article, and approved the final draft.
- Cristian G. Bologa conceived and designed the experiments, performed the experiments, analyzed the data, authored or reviewed drafts of the article, and approved the final draft.
- Anna Waller performed the experiments, authored or reviewed drafts of the article, and approved the final draft.
- Stephan C. Schürer performed the experiments, authored or reviewed drafts of the article, and approved the final draft.
- Dušica Vidović performed the experiments, authored or reviewed drafts of the article, and approved the final draft.
- Keith J. Kelleher performed the experiments, prepared figures and/or tables, authored or reviewed drafts of the article, and approved the final draft.
- Timothy K. Sheils performed the experiments, prepared figures and/or tables, authored or reviewed drafts of the article, and approved the final draft.
- Lars Juhl Jensen conceived and designed the experiments, performed the experiments, analyzed the data, authored or reviewed drafts of the article, and approved the final draft.
- Christophe G. Lambert conceived and designed the experiments, performed the experiments, analyzed the data, authored or reviewed drafts of the article, and approved the final draft.
- Tudor I. Oprea conceived and designed the experiments, performed the experiments, analyzed the data, authored or reviewed drafts of the article, and approved the final draft.
- Jeremy S. Edwards conceived and designed the experiments, performed the experiments, analyzed the data, authored or reviewed drafts of the article, and approved the final draft.

### Data Availability

The source code is not being submitted for review because it is a complex web application and the source code itself is not the subject of the manuscript, but rather the use of the resulting web application available at https://newdrugtargets.org, and the API at https://api.newdrugtargets.org. We make no references to specific lines of source code in the manuscript. The manuscript intentionally avoids technical discussion of the specific web technologies that were used to create the tool, instead focusing on explaining the theory behind this bioinformatics tool along with practical applications. We do have a public GitHub repository for the UI (https://github.com/unmtransinfo/tinx-ui) and also for the API (https://github.com/unmtransinfo/tinx-api), and these repository URLs are featured prominently in the main text of the submitted manuscript.

The TIN-X User Interface is available at GitHub and Zenodo:

-https://github.com/unmtransinfo/tinx-ui

-dccannon, & vmetzger09. (2024). unmtransinfo/tinx-ui: v3.0_tinx-ui (v3.0). Zenodo. https://doi.org/10.5281/zenodo.10690898.

The TIN-X API is available at GitHub and Zenodo:

-https://github.com/unmtransinfo/tinx-api

-dccannon, Justin Edwards, & vmetzger09. (2024). unmtransinfo/tinx-api: v3.0_tinx-api (v3.0). Zenodo. https://doi.org/10.5281/zenodo.10690905.

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
