# Peer review of "TIN-X version 3: update with expanded dataset and modernized architecture for enhanced illumination of understudied targets"

_PeerJ, doi:10.7717/peerj.17470_

## Round 0.1 · original submission · Minor Revisions

Thank you for your patience during the review process.

There was a fairly significant difference in opinions among reviewers. While reviewers 1 and 3 were supportive with only minor questions/revisions, reviewer 2 had more significant questions as to the degree of advancement from the original 2017 manuscript as well as other concerns about the data, website and APIs.

While there is a somewhat large difference in opinion, the broader consensus is that the paper can be accepted with revisions.

While some comments could make valuable additions, place primary focus in your revision on addressing reviewer comments related to comparison with similar or relatedtools. Similarly, responding to reviewer 2's concerns regarding explanations of the API/website would be valuable, along with more directly addressing where improvements will be expected to have an impact on the use or effectiveness of this tool.

Thank you again for your patience.

Reviewer 1 ·

Basic reporting

Authors have made novel contributions in updating the TIN-X tool. The manuscript is well written and explains how the updated tool performs the text mining based analysis and provides visualization of the association between proteins and diseases. In the current implementation, TIN-X supports exploration of data for G-protein coupled receptors, kinases, ion channels, and nuclear receptors. New features added to the tool are well explained and the User interface is user-friendly to access. Figures and tables are depicted clearly with overall workflow and the integrated tools like Pharos and Drug Central.
As a suggestion, in addition to the gene (target) disease associations, protein-protein interactions would be of interest to the users.
Although, TIN-X is novel and innovative for the target disease associations, comparison with similar tools that are related to the target disease prediction using literature can be included in the manuscript.

Experimental design

Upgraded TIN-X database with Amazon RDS and newly designed API are described in the paper. The tools  supports the browsing and navigating across proteins and diseases based on Importance and Novelty score. Illuminating the understudied targets in diseases such as Parkinson's diseases, is of much interest to the users.
Novelty and important scores provides the insights of target disease associations, incorporation of disease pathway information from the text will help understand the disease mechanism as well.

Validity of the findings

Authors have performed the case study on Parkinson's disease and TIN-X interface is used to explore understudied targets associated with the disease.
Evaluation of the performance of TIN-X with other similar tools would be of much interest for the users and helps in the validation of the tool. It might be interesting if authors could add up this in the current manuscript revision.

Additional comments

Inclusion of small RNA targets in addition to gene disease associations, will be an additional resource for the tool and help researchers enhance their knowledge of the disease targets.

Reviewer 2 ·

Basic reporting

The article does not seem to be self-contained. The major advancement over the 2017 publication of the same website appears to be an update to DISEASES database which it imports, integrates with the Target Central Resource Database, then displays with table and scatterplot widgets. The case studies it presents are weak and do not clearly connect to the update in the underlying database.

Experimental design

It's not clear what the scientific problem this update is trying to solve on top of the original 2017 publication of the same website. It makes minor UI updates, imports a new version of external resources, and presents various technical improvements that might be interesting, but are confusing / poorly documented / hard to use (such as the API and source code).

Validity of the findings

Due to the unclear stated goals of the manuscript, this is hard to comment on. The underlying data have explicitly not been provided and reproduction is not possible in its current state.

Additional comments

Overall
=====
It's not clear what the scientific problem this update paper is trying to solve on top of the original 2017 publication. The minor UI improvements do not seem to justify another scientific publication. Most of the improvements seem to be described as updates to the DISEASES database, which is not the nominal subject of this article. The case studies on Parkinson's disease is a relatively weak example, and does not seem specifically connected to updates in this manuscript.

The mentions of the integration with Pharos are also confusing and hard to follow. It's not clear what the context was about this integration nor if this is novel work presented as part of the paper. Further, it's not obvious how the figure about Pharos was made, or what the scientific utility is of such a view (from the text of the manuscript).

Backend/Data
===========
Further, this update lacks the "open science" flair that would make some of its content useful. Most importantly would be to make several different flavors of dumps of the processed data available in TSV, JSON, etc. to make it possible to access and download the dataset in bulk. Many of the functionalities of the website could be replicated with standard data science tools like pandas or visualization tools, making analysis much more flexible for users.

It's nice that https://github.com/unmtransinfo/tinx-api and https://github.com/unmtransinfo/tinx-api are mentioned in the paper, but they are not referenced from the website. More importantly, neither are adequately documented. Both should give more insight into what they are for, and _why_ someone would want to run this locally. Ideally, the steps in the readme of https://github.com/unmtransinfo/tinx-api would be fully automated inside a docker image that can be simply pulled from Docker Hub and significantly simplified. As it is, there are many difficult steps towards making this work that I was not able to perform. That being said, having a well-documented REST API is a much better substitute, or even better, making a TSV downloadable that can be worked with by any standard data science toolkit.

Web Site
=======
Why isn't information from underlying disease ontology propagated into https://newdrugtargets.org/?disease=DOID:0080194?

Inconsistent UI on https://newdrugtargets.org/?disease=DOID:0080194 - family is written as "(Unknown)" but DTO ID is blank.

"DTO ID:" is misleading, since what appears in this field is a Compact URI that includes a redundant prefix in it. Better to write "DTO CURIE:"

Pages are buggy - https://newdrugtargets.org/?disease=DOID:0080194 shows "1 of 0 publications"

Website doesn't appear to link to source code nor an explanation of the Python API that is referenced in the "About" modal. I was able to find a link to the REST API hidden in the paper text at https://api.newdrugtargets.org/. This should be linked in the website as well. However, https://api.newdrugtargets.org/ doesn't itself make sense, and you have to click again on "API Documentation" to actually get to the docs. The API documentation itself is pretty confusing and references a non-standard Python tool called "coreapi-cli". Somehow, the API actually doesn't document how to use itself outside of vendor lock-in to this tool. It would be much better for users to use a standard tool for documenting APIs such as OpenAPI, which generates neutral documentation about how to make GET, POST, or other requests using standard tools and syntax. Specifically, it is missing descriptions and examples that would be helpful for actually using the endpoints. For example, it's not clear what a publication ID is - does this mean a PubMed identifier? Should it be written as a number or a CURIE? Same thing with DOID and other identifiers.

It's unclear what the updates are to TIN-X besides using a new version of an external dataset and minor UI upgrades. The website has a section mentioning what's new, but many of these improvements such as switching underlying database management system, Python implementation, etc. don't affect the scientific bottom line. Compared to the figures in the last TIN-X paper in 2017 (https://academic.oup.com/bioinformatics/article/33/16/2601/3111842), it appears to look and work about the same.

·

Basic reporting

This paper describes the release of a new version of an innovative software released in 2017, the “ Target 79 Importance and Novelty eXplorer” (TIN-X). The authors provide good justification and background for the utilization of this software for hypothesis building and target selection, specifically by citing use-cases for the illumination of understudied drug targets.

Experimental design

The paper is concise, befitting an update to an already existing software, motivating the users to use the newer version by citing relevant results such as significant expansion of the used database to include full-text articles rather than just abstracts, and further improving the user interfaces to the tool by developing a REST API and improved visualizations.

Validity of the findings

While the update to the software and the improvements are sufficient as the paper stands, As a potential user of this software I would suggest including a global interactive visualization of the understudied proteome (perhaps in a graph/network style), where the understudied proteins can be highlighted in the context of the known interactome, and visualized within their functional clusters.

---

## Round 0.2 · accepted · Accept

Thank you for fully addressing the reviewer concerns and your manuscript is ready for publication. Congratulations again!